# Effect of Pressure and Temperature on CO_2_/CH_4_ Competitive Adsorption on Kaolinite by Monte Carlo Simulations

**DOI:** 10.3390/ma13122851

**Published:** 2020-06-25

**Authors:** Guanxian Kang, Bin Zhang, Tianhe Kang, Junqing Guo, Guofei Zhao

**Affiliations:** 1College of Safety and Emergency Management Engineering, Taiyuan University of Technology, Taiyuan 030024, China; kangguanxian0120@link.tyut.edu.cn; 2Key Laboratory of In-situ Property-improving Mining of Ministry of Education, Taiyuan University of Technology, Taiyuan 030024, China; kangtianhe@tyut.edu.cn (T.K.); guojunqing@tyut.edu.cn (J.G.); zhaoguofei@link.tyut.edu.cn (G.Z.)

**Keywords:** temperature, competitive adsorption, kaolinite, Monte Carlo simulations

## Abstract

The adsorption of CO_2_ and CO_2_/CH_4_ mixtures on kaolinite was calculated by grand canonical Monte Carlo (GCMC) simulations with different temperatures (283.15, 293.15, and 313.15 K) up to 40 MPa. The simulation results show that the adsorption amount of CO_2_ followed the Langmuir model and decreased with an increasing temperature. The excess adsorption of CO_2_ increased with an increasing pressure until the pressure reached 3 MPa and then decreased at different temperatures. The $S_{CO_{2}/CH_{4}}$ decreased logarithmically with increasing pressure, and the $S_{CO_{2}/CH_{4}}$ was lower with a higher temperature at the same pressure. The interaction energy between CO_2_ and kaolinite was much higher than that between CH_4_ and kaolinite at the same pressure. The interaction energy between the adsorbent and adsorbate was dominant, and that between CO_2_ and CO_2_ and between CH_4_ and CH_4_ accounted for less than 20% of the total interaction energy. The isothermal adsorption heat of CO_2_ was higher than that of CH_4_, indicating that the affinity of kaolinite to CO_2_ was higher than that of CH_4_. The strong adsorption sites of carbon dioxide on kaolinite were hydrogen, oxygen, and silicon atoms, respectively. CO_2_ was not only physically adsorbed on kaolinite, but also exhibited chemical adsorption. In gas-bearing reservoirs, a CO_2_ injection to displace CH_4_ and enhance CO_2_ sequestration and enhanced gas recovery (CS-EGR) should be implemented at a low temperature.

## 1. Introduction

As CO_2_ emissions are increasing, global warming is becoming an increasingly serious environmental problem [1,2]. In order to reduce CO_2_ in the atmosphere, the effective capture and sequestration of CO_2_ has received a greater amount of attention. Compared with activated carbons, zeolites, and metal organic frameworks (MOFs), clay minerals may be more suitable for CO_2_ capture and sequestration [3]. This is because a clay mineral is a natural adsorbent that is widely available in the soil and sedimentary environment and is cheap and easily available [4]. Furthermore, the adsorption of molecules on porous media, such as shale and coal, can be significantly affected by clay minerals which possess a large surface area [5,6,7]. Compared with CH_4_, CO_2_ can be preferentially adsorbed on shale and coal [8]. This means that CH_4_ can be displaced from a coal bed or shale gas reservoir by injecting CO_2_ into the gas-bearing reservoir to improve gas recovery (CS-EGR) [9]. Hence, a deeper understanding of the CO_2_/CH_4_ competitive adsorption behavior in clay will help optimize the CS-EGR technology.

Much of the literature refers to studying the adsorption behaviors of CH_4_ in clays [10]. There are also several experiments relating to CO_2_ adsorption on clays. Rother et al. [11] studied the CO_2_ adsorption on sub-single hydration layer montmorillonite clay by excess sorption and neutron diffraction. The results show that the maximum CO_2_ concentration was 0.15 g/cm^3^, after which the excess adsorption decreased linearly to zero, and the negative value increased with the increase in the CO_2_ bulk density. Alhwaige et al. [12] measured the adsorption ability of clay-reinforced bio-based chitosan polybenzoxazine nanocomposites for CO_2_. They found that the adsorption capacity of CO_2_ and the reversibility of adsorption–desorption were excellent, and the reversible adsorption capacity could reach 5.72 mmol/g. However, research on the competitive adsorption behavior of CO_2_/CH_4_ in clay is very limited.

In the past few years, molecular simulations have been executed to research CO_2_/CH_4_ competitive adsorption on clay minerals. Yang et al. [13] researched the adsorption behavior of CH_4_, CO_2_, and CH_4_/CO_2_ on Na-montmorillonite by grand canonical Monte Carlo (GCMC) simulations. The results indicate that the adsorption capacity of CO_2_ was higher than that of CH_4_. At y_CO2_ = 0.5, the selectivity of CO_2_/CH_4_ was generally in the range of 25.0–46.8. Jin and Firoozabadi [14] found that the adsorption of CH_4_ and CO_2_ is mainly dependent on the surface area of Na-montmorillonite. In addition, the adsorption of CO_2_ increased rapidly with the enhanced cation exchange under low pressure. They further studied the effect of the water content of montmorillonite on its adsorption of CH_4_ and CO_2_. The results show that water molecules are preferentially adsorbed on the surface of Na-montmorillonite at less than 10 MPa. CH_4_ and CO_2_ are adsorbed on the surface of water molecules to form a weak second adsorption layer. CO_2_ can exhibit multi-layer adsorption with an increased pressure, but CH_4_ cannot. The adsorption behavior of CH_4_ and CO_2_ has been studied for years, but few researchers have focused on the influence mechanism of a clay structure on the competitive adsorption between CH_4_ and CO_2_.

Based on the above analysis, in this investigation, we studied the adsorption of CO_2_ and CO_2_/CH_4_ mixtures on kaolinite by GCMC simulations with different temperatures (283.15, 293.15, and 313.15 K) up to 40 MPa. The interaction energies, isosteric heat of adsorption, and radial distribution function (RDF) were also analyzed. Kaolinite is one of the most abundant components in clay minerals [15]. As inorganic matter, due to the large specific surface area of kaolinite, it has a great significance for CO_2_ capture and CH_4_ production to study the interaction mechanism between kaolinite and CH_4_/CO_2_ [16]. In this research, we hope to clarify details of CO_2_/CH_4_ competing adsorption behavior accompanying the CS-EGR process.

## 2. Simulation Methods 

### 2.1. Models

The kaolinite layered configuration determined by Bish’s experiment [17] was directly used. Our research object was a 4 × 2 × 2 kaolinite (001) surface supercell model, and the detailed modeling processes and parameters were the same as in our previously published works [18,19]. An all-atom model was used to represent methane, where the C-H bond length was 0.109 nm and the C-H bond angle was 109°28’ [20,21]. A CO_2_ molecule was described as a three-center model (EPM2) and the C-O bond length was 0.1149 nm [22].

### 2.2. Interaction Potential Model

The interactions between CO_2_/CH_4_ and the kaolinite were simulated using the Dreiding force field [23,24]. The atomic charge and Lennard–Jones parameters were taken from Zhang et al. [18,19], Zhang et al. [25], and Harris and Yung [26], as listed in Table 1. We estimated the Coulomb interaction between the charges in the system utilizing Ewald’s summation method, the accuracy of which is 1 × 10^−5^ kcal mol^−1^. The van der Waals interaction was determined by utilizing an atomic base cutoff radius with a value of 0.8 nm [26]. A much larger vacuum space than L_X_ or Ly was placed along the Z direction in the simulation cell. The long-range electrostatic interactions and the slab geometry were accounted for by the three-dimensional Ewald summation considering the correction term [27,28].

### 2.3. Simulation Details

We calculated the adsorption of CO_2_ and CO_2_/CH_4_ mixtures on kaolinite by utilizing GCMC simulations, which have been widely used to solve adsorption problems. The isosteric heat of adsorption, interaction energies, and radial distribution function (RDF) with different temperatures (283.15, 293.15, and 313.15 K) and pressures of up to 40 MPa were analyzed.

Peng–Robinson’s state equation [29] was used to obtain the fugacity, which was the effective pressure of gas. For the calculation, the periodic boundary conditions and metropolis arithmetic rue [30] were used to accept or reject the generation, disappearance, translation, and rotation of methane molecules, depending on energy changes. The simulation generated 1 × 10^8^ configurations [31]. Half of the system was configured to maintain balance and the other was used for calculations. The sorption module of the Materials Studio software was employed in all GCMC simulations [32]. The absolute adsorption was fitted by the Langmuir model, which is expressed as
(1)$$n_{a}=\frac{N_{L}P}{\left(P_{L}+P\right)}\;$$

The Langmuir pressure (*P_L_*) represents the pressure at which the gas adsorption capacity is half of the maximum gas adsorption capacity. *P_L_* values are typically used to evaluate the methane affinity of absorbents and the feasibility of gas desorption under reservoir pressures, with lower *P_L_* values indicating that methane adsorption occurs more readily and that desorption is more difficult to achieve.

The relationship between excess adsorption and absolute adsorption was
(2)$$n_{e}=n_{a}-\upsilon \rho .$$

In this paper, the free pore volume of the kaolinite model was calculated by inserting the probe, utilizing the Atom Volumes and Surface tool in the Materials Studio software.

The adsorption selectivity in the binary mixtures can reflect the relative adsorption priority between CO_2_ and CH_4_, which is defined as [33].
(3)$$S_{CO_{2}/CH_{4}}=\frac{x_{CO_{2}}/x_{CH_{4}}}{y_{CO_{2}}/y_{CH_{4}}}.$$

It should be noted that when $S_{CO_{2}/CH_{4}}$ > 1, *CO_2_* is preferably adsorbed on the kaolinite, and a higher selectivity reflects that the adsorption capacity of CO_2_ relative to CH_4_ is greater.

The van der Waals energy and electrostatic energy together constitute the total interaction energy [18,34].

Direct electrostatic and higher-order interactions were neglected in the calculation of the total interaction energy. For a simple molecular model, the total interaction potential is underestimated by 5–15% [23,35].

The interaction energy was calculated according to
(4)$$E_{\mathrm{Interaction}}=E_{AB}-\left(E_{A}+E_{B}\right).$$

We used the Clausius Clapeyron equation to calculate the isothermal adsorption heat $q_{st}$, kJ/mol, which implied the information of energy release in the adsorption process [36].
(5)$$q_{st}=RT^{2}{\left[\frac{\partial \left(\ln P\right)}{\partial T}\right]}_{N}$$

The radial distribution function g(r) was used to calculate the relationship between the density variation of the guest molecule (CO_2_, CH_4_) and its distance to the surface of kaolinite [37].
(6)$$g_{ij}\left(r\right)=\frac{dN}{4\pi \rho_{j}r^{2}dr}$$

## 3. Results and Discussion

### 3.1. Validation

The model, force field, and interaction between CH_4_ and kaolinite were verified by experiments on lattice parameters, pore volumes, and adsorption isotherms in our previous work. Our simulation results were compared with the experimental results of Chen and Lu [38], which validated that the interaction between CO_2_ and kaolinite was reasonable.

Figure 1 shows the adsorption isotherms of CO_2_ of the simulation results and the experimental results at the same temperature of 298.15 K. The simulation results of GCMC are nearly consistent with the experimental results. The slight differences were due to the differences in samples and methods [9,39]. These differences were also found in the study of Xiong et al [40]. Therefore, the adsorption of CO_2_ and the CO_2_/CH_4_ mixture on kaolinite could be studied further using a model and force field. 

### 3.2. Single Component Adsorption

The single component adsorption of CH_4_ on kaolinite with different temperatures was studied in detail in our published paper [41]. Figure 2 displays the adsorption isotherms of CO_2_ at different temperatures. The molecule simulation study of Yang et al. [13] and Jin et al. [14] showed that the CO_2_ adsorption on montmorillonite clay. In terms of relative CO_2_ sorption capacity: montmorillonite > kaolinit. This may due to the fact that CO_2_ was adsorbed only on the external surface of kaolinite; however, adsorption also occurred in the interlayer space of montmorillonite, which had a larger interlayer distance than the size of a CO_2_ molecule. 

The results demonstrate that the adsorption capacity of CO_2_ increased with the increase in the adsorption equilibrium pressure, and increased rapidly at low pressure. The adsorption capacity of CO_2_ decreased as the temperature increased.

We studied the adsorption rate of CO_2_ by using the Langmuir equation to fit the simulation data (Figure 2) [6]. Table 2 shows the Langmuir constants. Figure 3 presents the temperature dependence of the CO_2_ maximum absolute amount adsorbed *N_L_*, cm^3^/g. It clearly shows that there is a negative linear relationship between *N_L_* and temperature, indicating that the CO_2_ maximum amount adsorbed decreased as the temperature increased.

The Langmuir pressure *P_L_* is the pressure at which the gas adsorption amount is *N_L_*/2. The *P_L_* value is usually applied to evaluate the molecule affinity of the adsorbent and the complexity of gas desorption. The Langmuir constant *P_L_* of CO_2_ adsorption on kaolinite was positively correlated with temperature, which indicated that with the increase in temperature, the adsorption of CO_2_ is more difficult and desorption becomes easier.

Figure 4 displays the excess adsorption of CO_2_ on kaolinite at temperatures of 283.15, 293.15, and 313.15 K. Excessive adsorption at different temperatures improves to a maximum (about 3 MPa) as the pressure increases, and then decreases at a higher pressure. 

Figure 5 shows the temperature dependence of CO_2_ excess adsorption. It can be seen that the CO_2_ excess adsorption amount decreased linearly with temperature, indicating that the adsorption capacity of CO_2_ adsorption decreases when the temperature increases.

### 3.3. Adsorption of CO_2_/CH_4_ Mixtures

Figure 6 shows the adsorption isotherms of the CO_2_/CH_4_ binary mixture at the temperatures of 283.15, 293.15, and 313.15 k, for pressures up to 40 MPa. As the pressure increases, the capacity of CO_2_ adsorbed increases rapidly, while the amount of CH_4_ adsorbed is more stable. The results show that CO_2_ is more preferentially adsorbed on the surface of kaolinite than CH_4_, because of the stronger interaction between CO_2_ and kaolinite, which was in accordance with the adsorption amount of pure CO_2_ and CH_4_.

The adsorption and separation properties of kaolinite for CO_2_ and CH_4_ were further evaluated by utilizing the adsorption selectivity $S_{CO_{2}/CH_{4}}$. The results are shown in Figure 7. It can be seen from the results that the selectivity is always larger than 3, which means that kaolinite has a high adsorption separation behavior for the CO_2_/CH_4_ mixture. When the pressure increases from 1 to 40 MPa, the $S_{CO_{2}/CH_{4}}$ decreases logarithmically and the $S_{CO_{2}/CH_{4}}$ is lower with a higher temperature at the same pressure. Hence, a CO_2_ injection in CH_4_-bearing reservoirs to displace CH_4_ and enhance the CS-EGR should be implemented at a low temperature.

### 3.4. Interaction Energies and Isosteric Heat Analysis during CH_4_/CO_2_ Adsorption

The interaction energy between the adsorbent and adsorbate and between the adsorbate and adsorbate [18,19] is shown in Figure 8 for the temperature of 293.15 K. Although each interaction energy decreased exponentially with the increase in pressure, the interaction energy between CO_2_ and kaolinite was much higher than that between CH_4_ and kaolinite at the same pressure.

This is the reason why the adsorption amount of CO_2_ is greater than that of CH4, and it is also the main reason why a CO_2_ injection can displace CH4 in CBM and shale reservoirs. In addition, the interaction energy between the adsorbent and adsorbate was dominant, and that between CO_2_ and CO_2_ and between CH_4_ and CH_4_ accounted for less than 20% of the total interaction energy, respectively, during the adsorption of the CO_2_/CH_4_ binary mixture.

Figure 9 shows the interaction energy curve for CO_2_ and kaolinite at different temperatures. Clearly, the temperature had an important influence on the interaction energy. The interaction energy between CO_2_ and kaolinite decreased with the increase in temperature and became less negative, which led to a decrease in the CO_2_ adsorption amount, consistent with Figure 3.

Figure 10 shows the calculation results of the isotherm adsorption heat of CO_2_ and CH_4_ on kaolinite at different temperatures. The isothermal adsorption heat of CO_2_ was higher than that of CH_4_, indicating that the affinity of kaolinite to CO_2_ was stronger than that of CH_4_. At low pressure, the isotherm adsorption heat of CO_2_ and CH_4_ on kaolinite increased with the increase in pressure. When the pressure was higher than 10 MP, it tended to be stable. With increasing temperature, the lower isotherm adsorption heat was not conducive to adsorption, in agreement with the results presented in Figure 4.

### 3.5. Radial Distribution Function

Figure 11 shows the RDFs of CO_2_ on kaolinite at a pressure of 5 MPa and the temperatures of 283.15, 293.15, and 313.15 K. The first and highest peak occurs at *r* = 0.33 Å, with values of about 1.92, 1.87, and 1.69 corresponding to the temperature of 283.15, 293.15, and 313.15 K. This means that the adsorption density of CO_2_ being found at *r* = 0.33 Å was 1.92, 1.87, and 1.69 times, respectively. The height of the first peak decreased with increasing temperature, indicating that the interaction between CO_2_ and kaolinite decreased with the increase in temperature. The results further confirm the influence of the temperature on the adsorption capacity of CO_2_ and the interaction energy between CO_2_ and kaolinite.

Figure 12 displays the RDFs between CO_2_ and different atoms of kaolinite at the pressure of 5 MPa and the temperature of 293.15 K. The first peaks of the RDFs between CO_2_ and H and CO_2_ and O appeared near *r* = 0.3 Å, which were sharp and intense, with peak values of 8.0 and 3.37, respectively. The first peak of the RDFs between CO_2_ and Si appeared near *r* = 2.0 Å, with a peak value of 3.25. In contrast, the close contact peak of CO_2_-Al was low, which indicated that the interaction between CO_2_ and Al was weak. We conclude that the strong adsorption sites of carbon dioxide on kaolinite are hydrogen, oxygen, and silicon atoms, respectively. It is worth noting that the non-bonding forces were mainly composed of hydrogen bonding forces and van der Waals forces, whose interaction ranges are 0.26–0.31 nm versus 0.31–0.50 nm. Hence, CO_2_ can form hydrogen bonds with hydrogen atoms in kaolinite. This means that CO_2_ is not only physically adsorbed on kaolinite, but also displays chemical adsorption, which was different from the adsorption of CH_4_ on kaolinite, which only demonstrated physical adsorption. In addition, the strength of the hydrogen bonding force is much greater than that of the van der Waals force. Therefore, the amount of CO_2_ adsorbed is much higher than that of CH_4_, which is consistent with the result of Figure 2.

## 4. Conclusions

Single component adsorption and CO_2_/CH_4_ mixtures on kaolinite associated with the CS-EGR process were simulated by GCMC up to 40 MPa with different temperatures (283.15, 293.15, and 313.15 K). The main findings are as follows:(1)The adsorption capacity of CO_2_ increased with an increase in the adsorption equilibrium pressure, and increased rapidly at low pressure. The adsorption amount of CO_2_ decreased as the temperature increased. Excessive adsorption at different temperatures increases to a maximum (about 3 MPa) as the pressure increases, and then decreases at a higher pressure;(2)The $S_{CO_{2}/CH_{4}}$ decreased logarithmically with increasing pressure, and the $S_{CO_{2}/CH_{4}}$ was lower with a higher temperature at the same pressure;(3)The interaction energy between the adsorbent and adsorbate was dominant, and that between CO_2_ and CO_2_ and between CH_4_ and CH_4_ accounted for less than 20% of the total interaction energy, respectively, during the adsorption of the CO_2_/CH_4_ binary mixture. The isothermal adsorption heat of CO_2_ was higher than that of CH_4_, indicating that the affinity of kaolinite to CO_2_ was higher than that of CH_4_;(4)The strong adsorption sites of CO_2_ on kaolinite are hydrogen, oxygen, and silicon atoms, respectively. CO_2_ is not only physically adsorbed on kaolinite, but also displays chemical adsorption;(5)By considering the results of adsorption, the capture and sequestration of CO_2_ and enhanced CS-EGR should be carried out at low temperatures.

## Figures and Tables

**Figure 1 materials-13-02851-f001:**
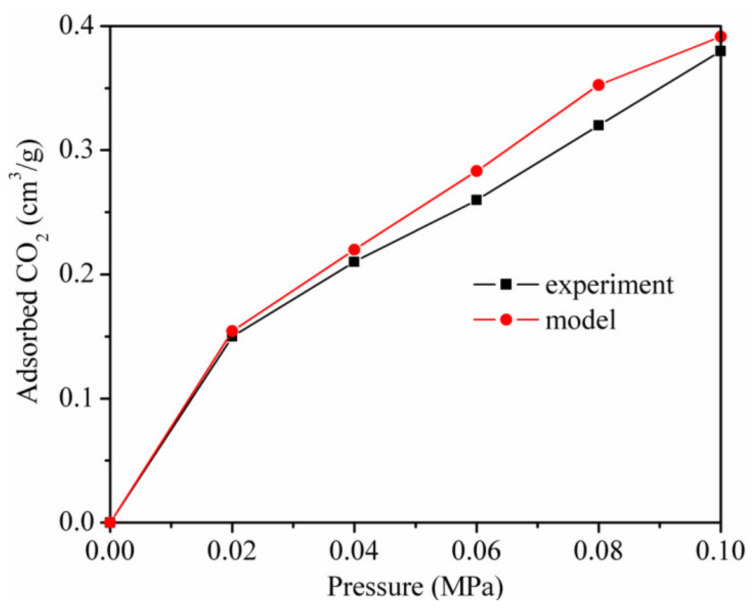
Adsorption isotherms of CO_2_ in terms of the simulation results and the experimental results at 298.15 K.

**Figure 2 materials-13-02851-f002:**
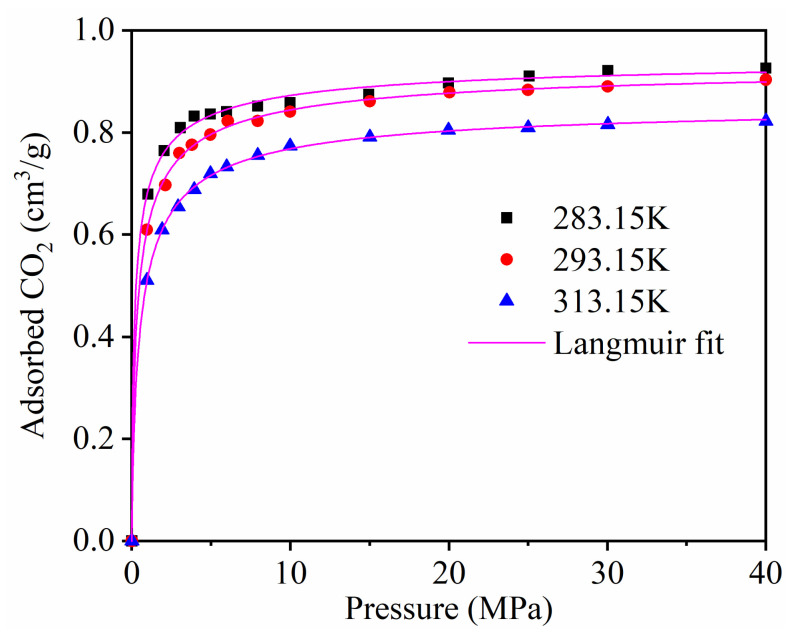
The adsorption isotherms of CO_2_ at different temperatures.

**Figure 3 materials-13-02851-f003:**
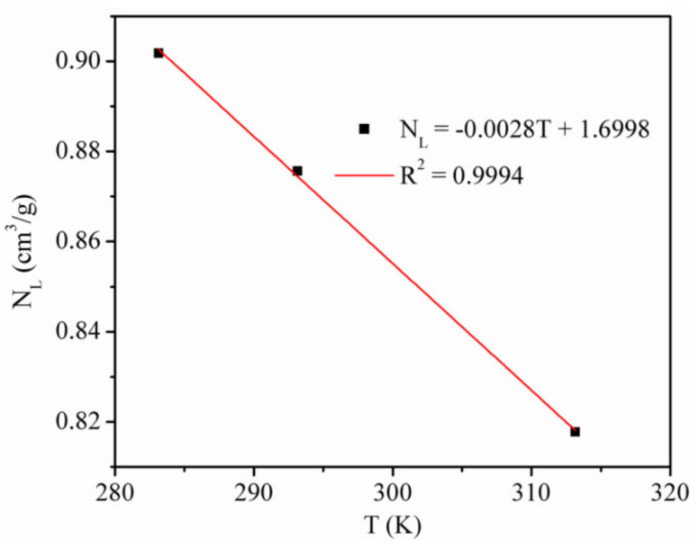
CO_2_ maximum absolute amount adsorbed at different temperatures.

**Figure 4 materials-13-02851-f004:**
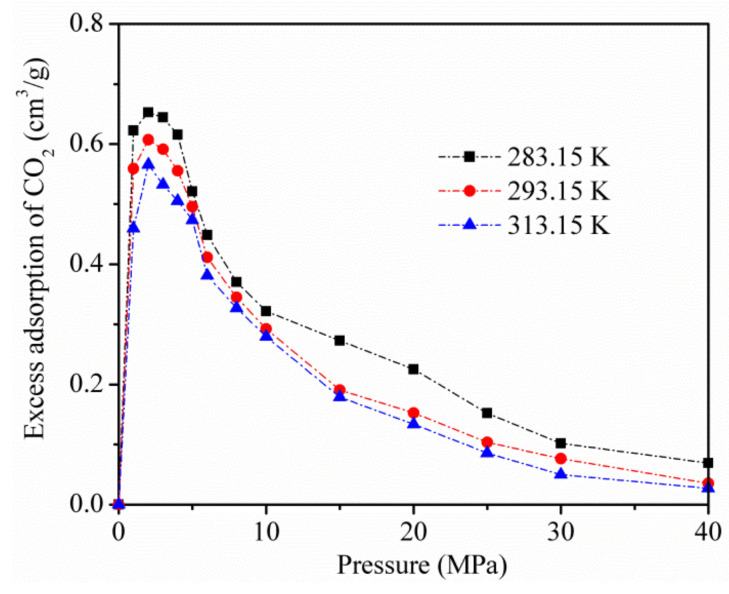
Excess adsorption of CO_2_ on kaolinite at temperatures of 283.15, 293.15, and 313.15 K.

**Figure 5 materials-13-02851-f005:**
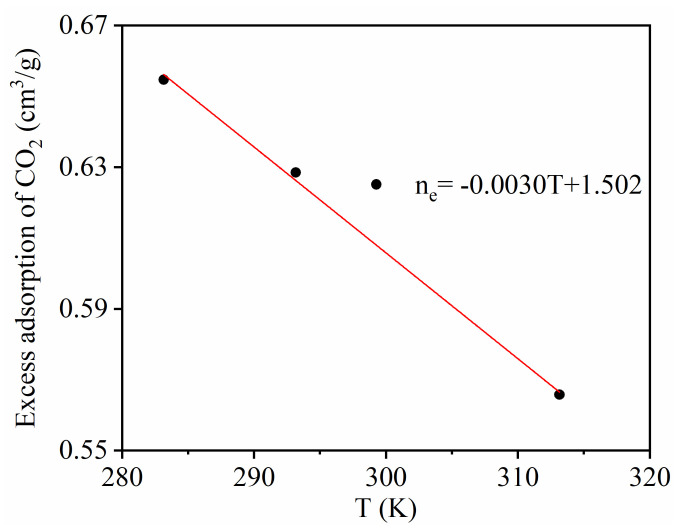
Temperature dependence of CO_2_ maximum excess adsorption.

**Figure 6 materials-13-02851-f006:**
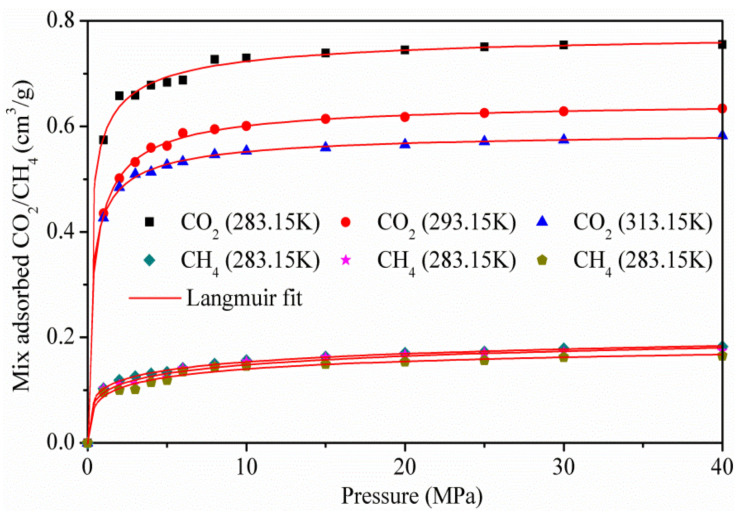
Isotherms of the CO_2_/CH_4_ mixture on kaolinite.

**Figure 7 materials-13-02851-f007:**
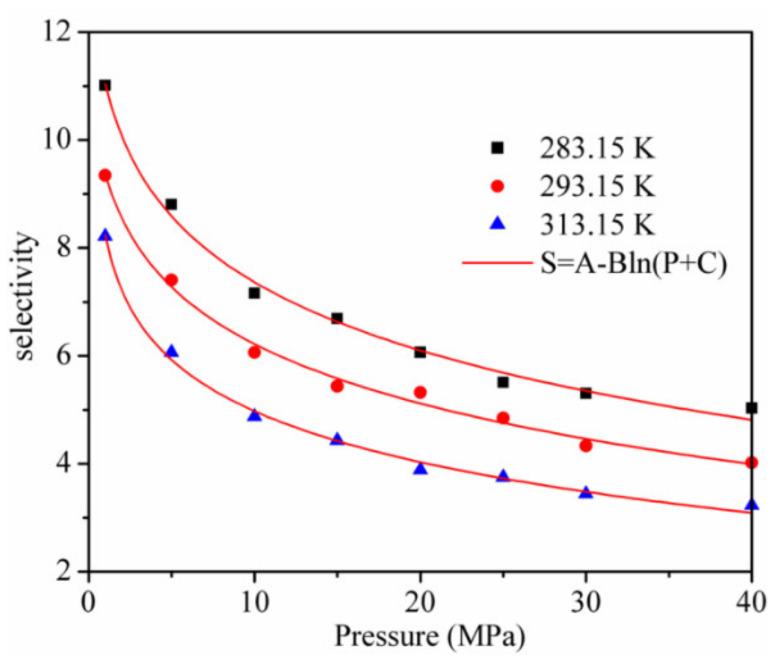
Selectivity of CO_2_/CH_4_ in kaolinite.

**Figure 8 materials-13-02851-f008:**
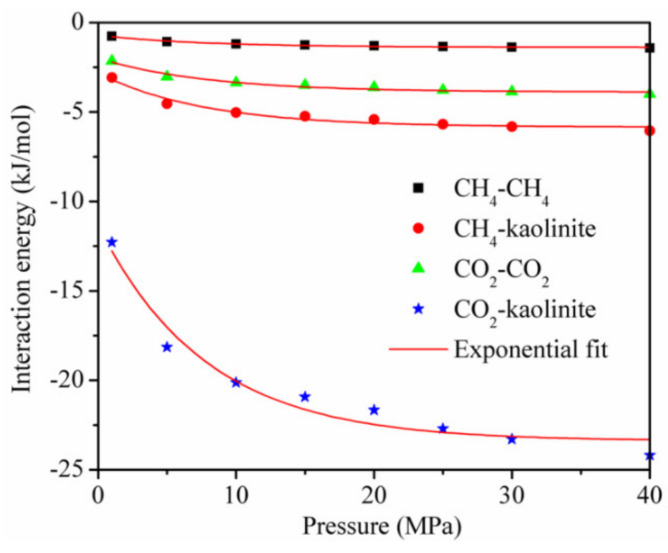
The interaction energy between the adsorbent and adsorbate and between the adsorbate and adsorbate at the temperature of 293.15 K.

**Figure 9 materials-13-02851-f009:**
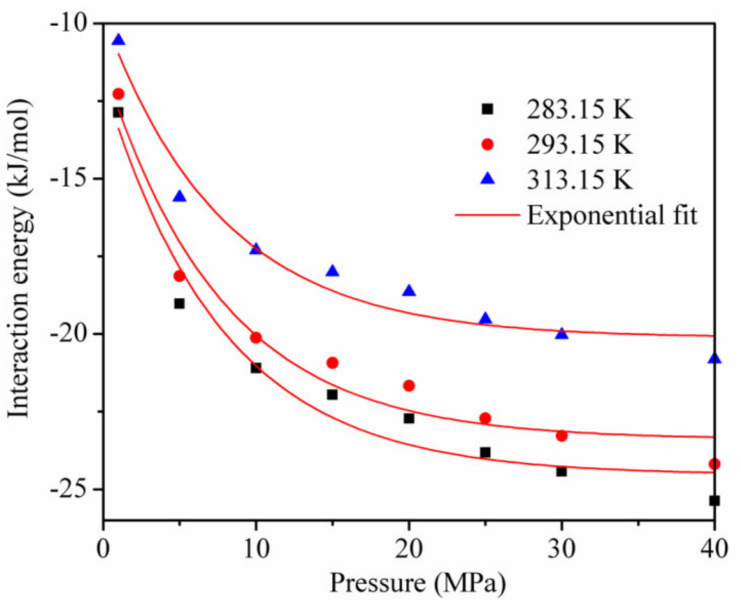
The interaction energy between kaolinite and CO_2_ at different temperatures.

**Figure 10 materials-13-02851-f010:**
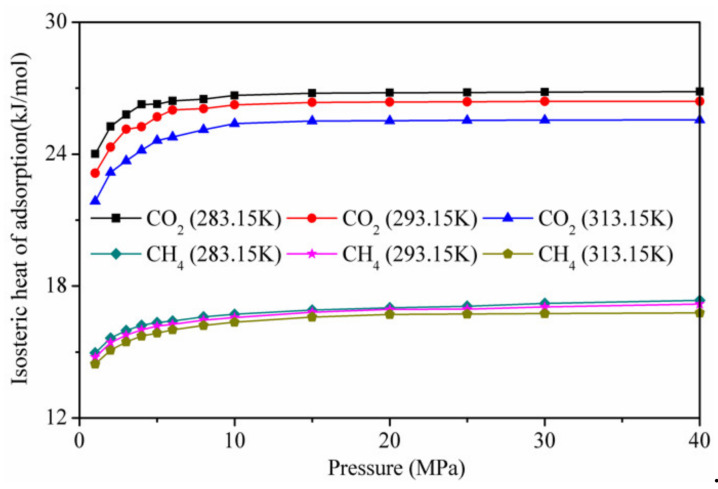
Isotherm adsorption heat of CO_2_ and CH_4_ on kaolinite at different temperatures.

**Figure 11 materials-13-02851-f011:**
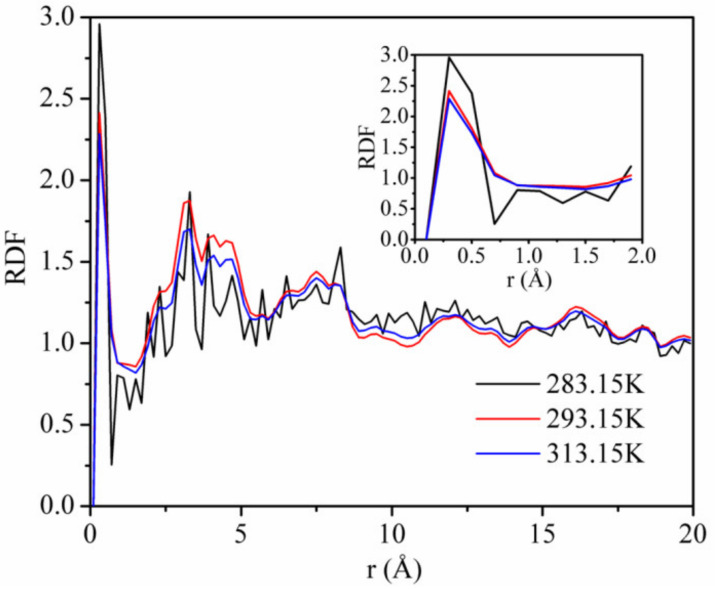
Radial distribution functions (RDFs) of CO_2_ on kaolinite at a pressure of 5 MPa and the different temperatures.

**Figure 12 materials-13-02851-f012:**
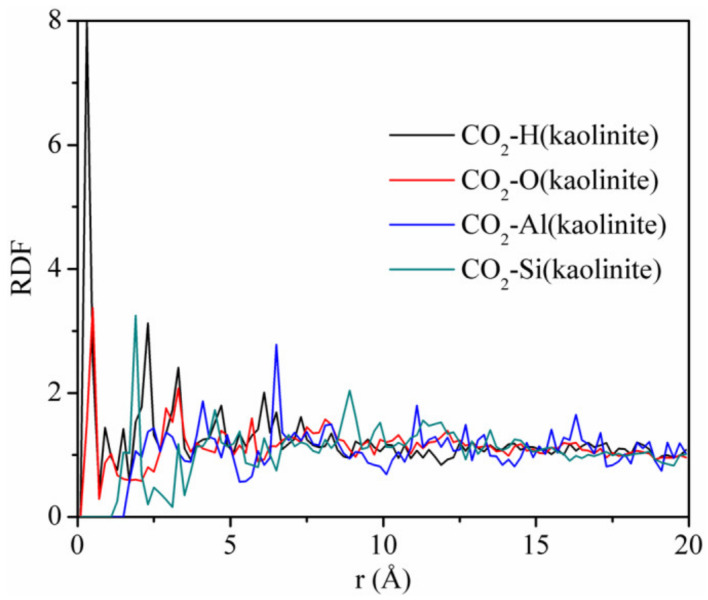
RDFs between CO_2_ and different atoms of kaolinite at the pressure of 5 MPa and the temperature of 293.15 K.

**Table 1 materials-13-02851-t001:** Lennard–Jones parameters and atomic charge.

Molecule	Element	σ/nm	$\left(\varepsilon /k_{B}\right)/K$	q/e
CO_2_	C	2.757		+0.6512
	O	3.033		−0.3256
CH_4_	C	3.184	0.06069	−0.1360
	H	2.963	0.06618	+0.0340
kaolinite	Si	0.400	0.20934	+1.1
	Al	0.420	0.20934	+1.45
	O(surface)	0.350	0.10467	−0.55
	O(apical)	0.350	0.10467	−0.75833
	O(hydroxyl)	0.350	0.10467	−0.68333
	H	0.1098	0.0544284	+0.2

**Table 2 materials-13-02851-t002:** Langmuir constants of adsorption of CO_2_ at temperatures of 283.15, 293.15, and 313.15 K.

Temperature	$N_{L}\;({\mathrm{cm}}^{3}/g)$	$P_{L}\;({\mathrm{Mpa}}^{-1})$	Correlation Coefficient R^2^
283.15 K	0.902	0.427	0.994
293.15 K	0.876	0.478	0.997
313.15 K	0.818	0.963	0.996

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
