# Peer review of "Effect of Pressure and Temperature on CO2/CH4 Competitive Adsorption on Kaolinite by Monte Carlo Simulations"

_materials, 2020, doi:10.3390/ma13122851_

Round 1
Reviewer 1 Report
The Authors report a theoretical study on the CO2 and CH4 adsorption capacity of kaolinite at three different temperature.
The manuscript contains interesting data, which, at the moment, have not properly valorised. In general, many English flaws are present, including very short sentences placed in the middle of a different discussion, confusion between past and present in the verbal time, some sentences not easily understandable. Also, the literature in the main text is reported sometimes with numbers (correct), and sometimes with the names and year in parenthesis. It is evident that transferring the manuscript from a previous format, part of the old style has not been properly converted.
Some examples representative:
- Line 11-12: verb missing
- Line 25-26: to add personal hopes about the manuscript in the abstract is unusual
- Line 29: increasing
- Line 30: “Recently, some scholars propose using CO2 capture and sequestration to reduce CO2 of the atmosphere.” It can be explained better
- Line 32: “This is due to clay mineral is a sort of cheap and easily available natural adsorbent widely existed in…..”
- Line 57: “(Jin & Firoozabadi, 2014).”
- Line 66: “Kaolinite is rich in clay minerals.”
- Line 67:”(Zhang et al., 2018c).”
- Line 80:” The atomic charge and Lennard-Jones parameters are taken from zhang et al.”
- Line 120: “results of Chen and Lu(Chen & Lu, 2015, which validated….”
Line 170: ”The results showed in Figure 6.”
- Line 195: “kaolinite became less negative, that is, it decreased with the”
And more.
From the scientific point of view, the study is not new, as very similar work has been reported, e.g. in the references 2 and 13. Nevertheless, in the discussion part, no comparison between previous similar studies and the current data is presented.
- Which is the relationship between the structure and the composition of the clay (montmorillonite or kaolinite) and the ability to adsorb CH4 and CO2?
- Why the choice of these specific three temperatures (283.15, 293.15 and 313.15 K) which are different from the ones reported in the more similar study (reference 2)? It is however possible a comparison?
- Line 82: “Ewald summation method, accuracy of which is 4.186×103 kJ mol-1.” It is this correct? and how it has been determined?
- Line 212: “This means that the probability of CO2 being found at r = 0.33Å was 1.92, 1.87 and 1.69 times, respectively” The probability with respect to?
- Line 254: “We expect that our research will help to the evolution of future optimized designs for the CS-EGR project for CBM and shale gas.” This is not a conclusion.
Reviewer 2 Report
This manuscript by Guanxian et al. aims to assess the competitive adsorption of CO2 and CH4 onto kaolinite under various pressure and temperature conditions. The topic is of high interest for the readership of Materials. Yet, even if the results and discussion section is interesting and in general sound, other sections deserve for major modifications as they are currently very hard to understand. A in-depth correction by a professional translator must be conducted. Therefore, i recommend to reconsider this manuscript after major modifications.
Specific comments are given below:
The abstract is very hard to follow with several purposeless sentences. It looks like a juxtaposition of non connected words. For example, the first sentence does not make any sense and should be rewritten. Please also avoid the repetitions (or copy/paste) between the abstract and the conclusion. Please define abbreviation in the abstract.
L29 "environmentAL"
L30 Ref 2 does not refer to global warming please modify.
L32-33 please reword this sentence
L34 Ref 4 is not relevant please modify
L34-35 why focus here on CH4 adsorption? This assumption is true for other molecules.
L51-53 what this study conclude on the selectivity CO2/CH4 this would be more consistent with the purpose of your work?
L57 please use a homogeneous format for reference citation
L57-58 "preferentially adsorbed on the surface of montmorillonite" please give a pressure range here, as Na-Mt is a swelling clay mineral
L66 Kaolinite is in fact a clay mineral
L66-67 Maybe, but here you study the interaction mechanism between CO2 and kaolinite? PLease explain? (the reference is also badly cited)
L120 Uncited reference, please revise
L123-Figure 1 what was the investigated temperature, 293 or 298?
L124-125 "methods similar to other literatures" This must be clarified.
Figure 2: Clearly the blue lines are not at all Langmuir fits but point to point connection, please revise
L138-139 This sentence is weird, especially if you don't any "other studies".
L140-141 These two sentences have the exact same meaning, please condense.
It would be good to indicate in Materials and Methods brief information on Langmuir equation
L150 Why focus here on CH4 only?
L157-158 You have to separately address the impact of pressure and temperature on the excess adsorption, it is currently too blur.
L228-229 For me, hydrogen bonds is not chemical adsorption, could you give any support to such classification?
L253-254 You have performed competitive adsorption, yet, the displacement of formerly adsorbed CH4 was not conducted, therefore it's hard based on your results to understand this sentence? Please explain?
Reviewer 3 Report
In this paper the authors study the effect of pressure and temperature on the carbon dioxide/methane competitive adsorption on kaolinite, by means of Monte Carlo simulation.
In general, this article is well structured, the concepts are clear and the methods used are adequate, however in some paragraphs it lacks depth. I suggest improving the introduction, both in form and content, to provide a more in-depth vision of the state of the art.
The abbreviations GCMC and CSEGR are indicated in the abstract. Although they are in common use and abundantly known to readers, it would be better to write them in full, on first use.
Lines 66-67: "It has great significance for shale gas ....": Authors should specify the reasons.
Lines 97-98: Authors should specify the software
Round 2
Reviewer 1 Report
All the issues have been properly addressed.
Reviewer 2 Report
The modifications performed on the manuscript are in accordance with the reviewer' suggestions, and therefore the revised version of the manuscript can be accepted as submitted.